# Impact of a Web-Based Nutrition Intervention on Eating Behaviors and Body Size Preoccupations among Adolescents

**DOI:** 10.3390/children10111736

**Published:** 2023-10-26

**Authors:** Manon Bordeleau, Maya Purcell, Véronique Provencher, Shirin Panahi, Raphaëlle Jacob, Natalie Alméras, Vicky Drapeau

**Affiliations:** 1Département d’Éducation Physique, Université Laval, Québec, QC G1V 0A6, Canada; manon.bordeleau.1@ulaval.ca (M.B.); maya.purcell@gmail.com (M.P.); 2Centre de Recherche de l’Institut Universitaire de Cardiologie et de Pneumologie de Québec-Université Laval (CRIUCPQ-UL), Québec, QC G1V 4G5, Canada; shirin.panahi.1@ulaval.ca (S.P.); natalie.almeras@criucpq.ulaval.ca (N.A.); 3Centre Nutrition, Santé et Société (NUTRISS), Institut sur la Nutrition et les Aliments Fonctionnels (INAF), Université Laval, Québec, QC G1V 0A6, Canada; veronique.provencher@fsaa.ulaval.ca; 4Centre de Recherche Interuniversitaire sur la Formation et Profession Enseignante (CRIFPE), Université de Montréal, Québec, QC H3C 3J7, Canada; 5École de Nutrition, Université Laval, Québec, QC G1V 0A6, Canada; 6Département de Kinésiologie, Faculté de Médecine, Université Laval, Québec, QC G1V 0A6, Canada; raphaelle.jacob.1@ulaval.ca; 7Department of Family Relations and Applied Nutrition, University of Guelph, Guelph, ON N1G 2W1, Canada

**Keywords:** adolescents, nutrition intervention, eating behavior traits, weight preoccupation, body size perception, body size dissatisfaction

## Abstract

We aimed to evaluate the impact of a web-based school nutrition intervention on eating behavior traits, body weight concern, body size perception and body size dissatisfaction in adolescents. Ten classes of secondary students in Canada (13.6 ± 0.8 years) were randomized into an intervention (*n* = 162 students) or control group (*n* = 75 students). Adolescents in the intervention, conducted between 2011 and 2013, participated in an online nutrition challenge to increase their consumption of vegetables, fruits and dairy products using a web-based platform over six weeks. Measurements were taken at baseline (PRE) and post-intervention (POST). No significant negative changes were observed between the intervention and control groups for eating behavior traits, body weight concern, body size perception and dissatisfaction. However, results suggest a trend for a positive effect of the intervention on susceptibility to hunger in boys (group × time interaction, *p* = 0.10). Specifically, boys experienced a reduction in their susceptibility to hunger in response to the intervention (PRE: 6.1 ± 3.8, POST: 4.8 ± 3.7, *p* = 0.009). An intervention aimed at improving the eating habits of adolescents did not negatively influence body size preoccupations. In response to the intervention, boys tended to show a lower susceptibility to hunger, which might help them to prevent overeating and adopt healthy eating habits.

## 1. Introduction

The prevalence of overweight and obesity in adolescents has prompted increased dietary and lifestyle strategies to prevent obesity or even reverse the condition [1]. In parallel, large community surveys have highlighted increased body weight preoccupation in youth of all body weight statuses [2]. Transversal and londitudinal studies have also observed an association between a negative body image and unhealthy eating behaviors in adolescents [3,4]. Perceived expectation to be thin and thin-ideal internalization may be considered risk factors for body weight concern (referring to worries or anxieties regarding one’s weight), body size dissatisfaction (reflecting discontentment with one’s body size) and propensity towards engaging in weight loss diets [5,6]. Therefore, some adolescents may be more motivated to participate in health promotion efforts if they have greater worries about their body weight. Consequently, it is essential to ensure that the implementation of such programs does not inadvertently contribute to negative health implications and psychosocial repercussions.

Few studies have evaluated the influence of healthy lifestyle promotion interventions on adolescents’ psychological health, including body weight concern, body size perception (referring to how individuals perceive their own body size), body size dissatisfaction and eating behavior traits [7]. Intervention effectiveness is typically based on physical or behavioral outcomes such as changes in body weight or diet; however, possible weight-related psychological effects are rarely assessed [1,8]. This is important since excessive body weight concern, body size overestimation and body size dissatisfaction can lead to increased dietary restriction and unhealthy dieting strategies, which have been linked to later-life weight gain [9,10]. Moreover, eating behavior traits, including higher levels of cognitive restraint (i.e conscious and deliberate efforts to control food intake), disinhibition (i.e., overconsumption due to a loss of control) and susceptibility to hunger (i.e., perceived hunger), were also found to be significant predictors of overweight/obesity and weight gain [11,12].

Furthermore, adolescence is a transitional period which involves the acquisition of healthy habits [13] that may continue into adulthood [14]. Adolescents are more vulnerable to developing weight and eating behavior problems [15]. Dieting and body weight concerns are more common in girls compared to boys [16] and in adolescents living with overweight or obesity (OW/OB) than in normal-weight adolescents [17]. Girls also tend to respond more positively to interventions that promote a healthy lifestyle than boys due to their greater effectiveness in achieving the intervention’s goals [18,19]. This aligns with the hypothesis that girls demonstrate higher motivation to adhere to the intervention than boys due to greater sociocultural pressure for girls to be thin [18].

Our research group has developed a web-based nutrition intervention, called Team Nutriathlon, which has been shown to be an effective and innovative nutrition intervention to promote healthy eating among youth [20,21]. This intervention is a 6-8-week nutrition challenge that encourages the consumption of a variety of vegetables, fruits and dairy products. It includes critical components of the self-determination theory [22] and behavioral change techniques [23], such as goal setting, self-monitoring, identification of barriers and solutions and social support, in order to influence youth’s behaviors positively. Our previous data have shown that this program positively impacted vegetable, fruit and dairy product consumption in children [20] and in adolescents in the short term [21]. However, the broader effects of this intervention on eating behavior traits and body weight have not yet been evaluated.

This study aimed to examine the impact of a web-based school nutrition intervention on eating behavior traits, body weight concern, body size perception and dissatisfaction in adolescents. A secondary objective was to investigate if sex and body weight status influence the response to the intervention. Considering the positive modalities of the intervention, we hypothesized that this web-based nutrition intervention will not influence psychological weight-related variables in adolescents, even in more susceptible adolescents such as girls and those living with OW/OB.

## 2. Materials and Methods

### 2.1. Participants

High schools across the Québec City region were recruited from 2011 to 2013 on a voluntary basis via email invitations. To participate in this study, interested teachers were required to meet the study’s inclusion criteria: teaching 1st- and 2nd-year high school classes corresponding to youth aged 13 and 14 years old. This randomized, clustered intervention included ten classes randomized to either the intervention, i.e., Team Nutriathlon (*n* = 6), or the control group (*n* = 4), which followed the regular school curriculum (Figure 1). Throughout the intervention, the participating schools adhered to the Quebec Secondary School Curriculum, referred to as the “Programme de formation de l’école québécoise.” Within this curriculum, there is no dedicated intervention designed to promote healthy eating habits or foster positive body image [24].

Randomization was carried out within the same school and during the same season for both the intervention and control groups to better control the environmental factors. The two public schools chosen for this study exhibited a medium-to-high socioeconomic status, as determined by the Deprivation Index established by the Ministry of Education of Québec [25].

Before the study, the research professional (coordinator) introduced and explained the project to the adolescents in their respective classes using the assent form. Subsequently, informed consent forms were sent home, and parents or legal guardians were encouraged to indicate their acceptance or refusal for their adolescent to enroll in the study (*n* = 282). Participants meeting the main inclusion criteria for the study had to have internet access. The present analyses included participants with complete data for the dependent variables. In contrast, those with missing zBMI data were excluded from the study (Figure 1). All adolescents with data for the zBMI completed the pre/post study assessments for the dependent variables.

### 2.2. Data Collection

#### Team Nutriathlon Intervention

At the beginning of the study, before the intervention, an initial education session was conducted in class for all participants by teachers, with the assistance of a study coordinator. This session aimed to familiarize them with the concept of serving sizes and use of the web-based platform. Following this session, all students were requested to document their daily consumption of vegetables/fruits and dairy products as a baseline measure. Additionally, they were asked to complete a questionnaire in paper format to collect data on eating behavior traits, body weight concern, body size perception and dissatisfaction (PRE, week 0).

After the baseline data collection, Team Nutriathlon, an online school nutrition intervention aimed at improving diet quality by increasing and adding diversity to their consumption of vegetables/fruits and dairy products, was explained to the intervention group (implementation phase). During this intervention, high school participants were motivated to achieve both team and individual goals targeting the quantity and variety of vegetables/fruits and dairy product consumption during six weeks. These goals were based on Canada’s Food Guide recommendations at the time of the study [26]. Over the six-week period, from Monday to Friday, participants in the intervention group used the Team Nutriathlon web-based platform to log their daily consumption of vegetables, fruits and dairy products. The intervention included three follow-ups or regulation periods (every two weeks) led by a dietitian or research professional from the research team, representing the intervention’s core. During the regulation periods, participants were asked to analyze their consumption over the last two weeks and the performance of their strategies. They had to find new strategies to help them reach the goals of the intervention. Following the intervention, all participants from both the control and intervention groups were requested to complete once again the same questionnaires they had at the beginning to gather data on eating behavior traits, body weight concern, body size perception and dissatisfaction (POST, week 7). A comprehensive design of this intervention is explained elsewhere [21].

### 2.3. Anthropometric Measures

The research professional and trained dietitians (*n* = 2) measured adolescents’ height and weight at school according to standardized methods [27] at baseline (PRE, week 0) and after the intervention phase (POST, week 7). Measurements were conducted privately for each participant. During the first week of the intervention, body mass index (BMI) was calculated from height and weight (kg/m^2^) and converted to a standardized z-score based on the 2007 WHO growth reference charts for ages 5–19 years, considering both age and sex [28]. These scores were employed to categorize the weight status of adolescents as underweight (zBMI < −2 standard deviation), normal weight (1 > zBMI ≥ −2), overweight (2 > zBMI ≥ 1) or obese (zBMI ≥ 2) [28]. Anthropometric measurements were retaken after the intervention (POST, week 7).

### 2.4. Eating Behavior Traits, Body Weight Concerns and Body Image Questionnaires

The French version [29] of the Three-Factor Eating Questionnaire (TFEQ) [11] was used to assess three dimensions of eating behavior traits, namely cognitive restraint, disinhibition and susceptibility to hunger. This questionnaire consists of 51 items, with 36 items presented in a true or false format, coded as 0 or 1. The remaining 15 items are assessed using a 4- or 6-point scale, which is coded as 0 or 1. Higher scores on each scale and subscale indicate higher levels of the respective eating behavior traits [11].

Body weight concern was measured with two questions from the Social and Health Survey [30] regarding weight preoccupation and desire for body weight modification. Body weight preoccupation was assessed with the question: “Do you consider that you are preoccupied with your physical appearance and/or body weight?” A positive response (“yes”) was coded as weight preoccupied, and a negative response (both “no” and “I do not know”) was coded as unpreoccupied. The following question assessed desires for body weight modification: “Among the following choices, which would you desire regarding your weight: to lose weight, to gain weight, to keep the same weight, or I do not worry about my body weight.” Those who responded as not being preoccupied with weight and desiring to keep the same weight were coded as “no desired weight change,” whereas the other responses were coded as “desired weight loss” or “desired weight gain”.

Body size perception and dissatisfaction were evaluated using the Collins Figure Rating Scale [31], where figures were numbered from one to seven, with the lowest representing an underweight figure and the highest representing an individual living with obesity. Adolescents identified their perceived actual body size followed by their desired body size. Following the method described by Maximova et al., we standardized the Collins figure scale for both perceived and desired body size, assigning a corresponding z-score (−3, −2, −1, 0, 1, 2, 3) to each figure [32]. Consequently, each figure was associated with weight status in a manner that figures with z-scores of −1, 0 and 1 representing a normal body weight. In contrast, z-scores of 2 and 3 standard deviations above and below the mean were associated with the overweight or obese and underweight classifications, respectively.

Based on the method described by Bordeleau et al. [33], the body size perception score was calculated as the difference between perceived actual body size and actual body size (zBMI). Adolescents were categorized into three groups according to their body size perception score: underestimators (body size perception score < −0.5); accurate estimators (−0.5 ≤ body size perception ≤ 0.5); and overestimators (body size perception score > 0.5). It should be noted that the misperception group comprises both underestimators and overestimators of their actual body size. Likewise, subtracting the desired body size score from the perceived body size score also resulted in three groups for the body dissatisfaction variable: desire to reduce body size (body size dissatisfaction score > 0.5); satisfaction with body size (−0.5 ≤ body size dissatisfaction score ≤ 0.5); and desire to increase body size (body size dissatisfaction score < −0.5).

### 2.5. Statistical Analyses

As represented in the flowchart (Figure 1), data from 237 adolescents (89 boys and 148 girls) were retained for analysis. Then, for statistical power purposes, the underweight group (*n* = 33) was merged with the normal weight group (*n* = 131) for a total of *n* = 164 participants classified in the UW/NW group. Likewise, participants with overweight (*n* = 51) were merged with participants living with obesity (*n* = 22) for a total of *n* = 73 participants in the OW/OB group.

Baseline means for continuous variables were compared between the experimental groups using Student’s *t*-tests. Chi-squared (χ^2^) tests were used to assess baseline differences in the prevalence of categorical variables. Changes in continuous variables in response to the intervention were assessed using mixed models for repeated measures. Generalized linear mixed models were used to examine changes in binary variables in response to the interventions. In both mixed linear models and generalized linear mixed models, group, time and their interaction were treated as fixed effects, and participants and classes were treated as random effects. In the case of a significant group-by-time interaction, the Tukey–Kramer post hoc test was used to identify within- and/or between-group differences. In the whole-group models, adjustments were made for age, sex and body weight status. In models stratified by sex, adjustments were made for age and body weight status, while in models stratified by body weight status, adjustments were made for age and sex. Statistical significance was set at a *p*-value ≤ 0.05. All analyses were performed with SAS OnDemand for Academics (Cary, NC, USA).

## 3. Results

### 3.1. Baseline Participant Characteristics

The mean age of the 237 adolescents was 13.6 ± 0.8 years, with 69% (*n* = 164) of them considered UW/NW (63% of boys, 73% of girls), and 31% (*n* = 73) living with OW/OB (37% of boys, 27% of girls). The baseline characteristics of adolescents in the intervention and the control groups are presented in Table 1. The only notable difference between these groups was that participants in the intervention group were slightly older than those in the control group (*p* < 0.05) and had a lower prevalence of weight preoccupation (*p* = 0.05).

### 3.2. Eating Behavior Traits

At baseline, there was no significant difference between the mean values of the control and intervention groups in any eating behavior (Table 1). When considering the entire group, without distinction by sex or group status, no significant group-by-time interaction was observed for cognitive restraint, disinhibition and susceptibility to hunger (*p* = 0.93; *p* = 0.23 and *p* = 0.12; respectively). There were no statistically significant differences in cognitive restraint or disinhibition before and after the intervention between the control and intervention groups, for both boys and girls (Appendix A). However, among boys, there was a trend for a group-by-time interaction (*p* = 0.10), suggesting a potential effect of time on decreasing susceptibility to hunger in response to the intervention (PRE: 6.1 ± 3.8, POST: 4.8 ± 3.7, *p* = 0.009). Post hoc analyses revealed a significant reduction in susceptibility to hunger among boys in the intervention group pre-intervention and post-intervention (*p* = 0.01). In contrast, no significant differences were observed in susceptibility to hunger between the PRE and POST-periods in the control group (*p* = 0.98). After stratifying by weight status, no significant group-by-time interactions were found in any of the eating behavior traits (Appendix A).

### 3.3. Body Weight Concern

After six weeks, there was no significant difference in weight preoccupation (Table 2). Moreover, there were no significant group-by-time interactions observed in the overall sample or when stratified by sex or body weight status. Similarly, when examining the desire for body weight change, no significant group-by-time interactions were found in the overall sample or when stratified by sex or body weight status (Table 2).

### 3.4. Body Size Perception and Body Size Dissatisfaction

At baseline, 86% of adolescents underestimated their body size, 12% had an accurate perception, and only 2% overestimated their body size. Moreover, body size misperception was more frequent among girls compared to boys (92% vs. 80%; *p* = 0.007) and in adolescents living with OW/OB compared to those in the UW/NW group (97% vs. 83%; *p* = 0.002). In the overall sample, as well as when stratified by sex or body weight status, no significant group-by-time interaction was observed (Table 3).

Regarding body size dissatisfaction, at baseline, 45% of the sample indicated being satisfied with their body size, while 38% shared the desire to reduce their body size and 17% to increase it. Adolescent girls were more inclined to express a desire to reduce their body size compared to boys (44% vs. 29%, respectively). No significant group-by-time interaction was noticed in the prevalence of body size dissatisfaction in any group (Table 3).

## 4. Discussion

This study evaluated the impact of a web-based school nutrition intervention, i.e., Team Nutriathlon, on changes in eating behavior traits, body weight concern and body size perception and dissatisfaction in adolescents, a population which needs to improve their eating habits but who are also vulnerable to weight-related psychological issues. We have previously shown that this intervention positively impacted the consumption of vegetables/fruits and dairy products among these adolescents, at least in the short term [21]. The present results add to the overall evaluation of this intervention and suggest that a school-based nutrition intervention challenge aimed at enhancing the quantity and variety in the consumption of vegetables/fruits and dairy products did not negatively impact eating behavior traits, body weight concern, body size perception and dissatisfaction in adolescents. It is crucial to measure these psychosocial variables, even in interventions that primarily target eating habits and may not directly address body image concerns in this more vulnerable population.

This school-based nutrition intervention did not negatively influence eating behavior traits. The non-significant impact on eating behavior traits may be attributed to the relatively short duration of our intervention, which might not have provided ample time for significant changes to be observed in these complex behaviors. The fact that the intervention did not directly focus on eating behaviors (more on vegetables/fruits and dairy product consumption) could also partly explain these non-significant results.

To the best of our knowledge, no previous research has examined this topic, making these findings novel and unique in the current literature. Moreover, the observation that boys who participated in Team Nutriathlon tended to decrease their susceptibility to hunger in response to the intervention aligns with previous research indicating that the consumption of vegetables/fruits and dairy products may promote satiety, which could potentially explain the decrease in susceptibility to hunger [34,35]. Notably, having a lower susceptibility to hunger is beneficial, as studies have shown that a higher susceptibility to hunger is a significant predictor of overweight/obesity and weight gain [12]. These findings are also consistent with previous studies conducted by Barnes and Kristeller (2016), who investigated the effects of a Mindfulness-Based Eating Awareness Training program for adolescents [36]. The workshops of this program address stress-related issues in adolescent physical and emotional health, targeting improved eating behavior by enhancing eating awareness. The program also promotes flexible changes in food choices, offers coping tools like relaxation techniques and encourages increased physical activity. The results indicated no effect on weight preoccupation and concerns about dieting. Still, interestingly, the intervention group showed a trend toward a greater decrease in perceived susceptibility for hunger compared to the control group in boys [36]. Given that this effect may prevent overconsumption/favor a healthy body weight trajectory [34], these findings support the relevance of interventions promoting healthy eating habits.

The present study also showed that, from baseline, most students did not accurately perceive their body size (i.e., perceived figure not matching with the one associated with their zBMI). Furthermore, approximately half of the students desired to be thinner or bigger. Despite a significant proportion of adolescents with misperception and a desire to be thinner, the findings of the present study suggest that the nutrition intervention did not influence body size perception and dissatisfaction, regardless of sex and body weight status. Such findings are important because previous research has demonstrated that adolescents who overestimate their body size and/or experience body size dissatisfaction are more inclined to adopt unhealthy behaviors such as skipping breakfast and/or dinner or to adopt healthy behaviors but not for good reasons, i.e., increasing their consumption of vegetables and fruits to control their weight [9,37]. Of note, in the present study, even if adolescents increased their consumption of vegetables/fruits in response to the intervention [21], there was no association between body dissatisfaction and changes in vegetable/fruit consumption (Appendix A). These findings may be attributed to the complex interplay of various factors influencing body image concerns, including societal norms and peer influences. Achieving significant changes in this regard may require a longer duration for the intervention. These results suggest that encouraging healthy eating habits does not harm how adolescents perceive their body size, at least within the context of a short-duration intervention.

The results of this study highlight the potential of web-based interventions in promoting healthy eating habits in adolescents without negatively affecting eating behaviors, body weight concerns and body image. Technology can help make the intervention more interactive and engaging for adolescents, increasing their motivation to participate and adopt healthy eating habits. The finding that the intervention did not negatively influence body size preoccupations indicates that it is possible to promote healthy eating habits safely in this population even with web-based intervention.

Strengths of this research include its comprehensive evaluation of a web-based school nutrition intervention, Team Nutriathlon, on a range of critical factors in adolescents’ well-being. Additionally, this study addresses a significant gap in the literature by exploring the intervention’s effects on psychosocial variables such as eating behavior traits, body weight concern and body size perception and dissatisfaction. These results contribute to a more holistic understanding of the intervention’s outcomes and its implications for adolescent health.

This study is not without limitations. First, it has been reported that Collins’ silhouettes do not present an adequate number of possible silhouettes, making it difficult for adolescents to accurately identify their body type [38]. No silhouettes represent a muscular build for boys or girls. Thus, it is difficult to differentiate between adolescents wishing to gain muscle mass versus those desiring to gain body fat. Additionally, due to the limited number of respondents in the underweight category, we combined underweight and normal weight respondents into one group, which could potentially impact the precision of our results. Finally, this study did not assess the long-term impact of the intervention on eating behaviors and body image. Subsequent investigations should evaluate the effectiveness of web-based interventions in promoting healthy eating habits among adolescents over an extended period.

## 5. Conclusions

A web-based nutrition intervention aimed at increasing adolescents’ vegetable/fruit and dairy product consumption and variety did not negatively influence eating behaviors traits, body weight concern, body size perception or dissatisfaction. This study suggests that this type of intervention could reduce susceptibility to hunger among adolescent boys. Finally, this study highlighted the importance of assessing eating behaviors and body image concerns during all nutrition and obesity prevention interventions in youth to understand if and how these interventions influence psychological and behavioral factors in this age group. Future research should examine the impact of a long-term intervention on these psychological variables.

## Figures and Tables

**Figure 1 children-10-01736-f001:**
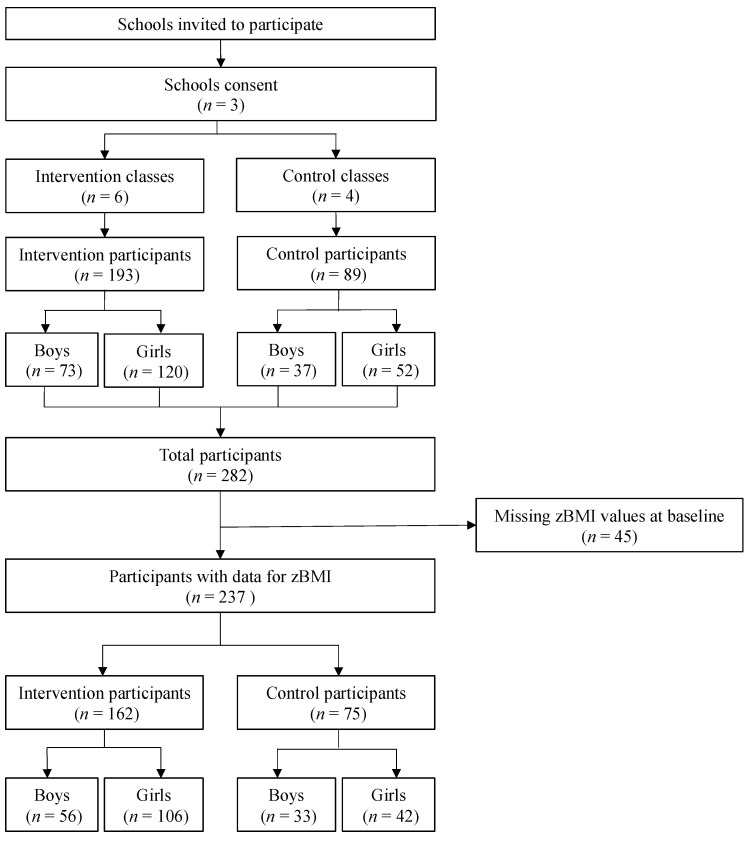
Flowchart of the study selection process.

**Table 1 children-10-01736-t001:** Baseline characteristics of participants in the intervention and the control groups.

	Intervention Group (*n* = 162)	Control Group (*n* = 75)	*p*-Value
Boys/Girls (%)	35/65	44/56	0.16
Age (y)	13.7 ± 0.8 (12.0, 16.0)	13.4 ± 0.8 (12.0, 15.0)	<0.05
zBMI	0.4 ± 1.2 (−2.68, 3.20)	0.4 ± 1.2 (−2.19, 3.01)	0.70
Body weight status			
Underweight/Normal weight/Overweight/Obese (%)	69/31	71/29	0.74
Underweight/Normal weight zBMI	−0.3 ± 0.9 (−2.68, 0.99)	−0.1 ± 0.8 (−2.19, 0.98)	0.72
Overweight/obese zBMI	1.7 ± 0.6 (1.01, 3.20)	1.7 ± 0.6 (1.04, 3.01)	0.99
Weight preoccupation (%)	42	56	0.05
Desire for body weight change (%)	85	84	0.81
Body size misperception (%)	86	91	0.29
Body size dissatisfaction (%)	55	56	0.88
Cognitive restraint	6.6 ± 4.3 (0, 21)	6.7 ± 3.9 (0, 17)	0.66
Disinhibition	4.7 ± 2.5 (0, 12)	4.7 ± 2.5 (0, 11)	0.77
Susceptibility to hunger	5.4 ± 3.6 (0, 14)	4.7 ± 3.6 (0, 14)	0.29

Data are presented as percentage or means ± SD (min, max). Cognitive restraint, disinhibition and susceptibility to hunger were adjusted for age, sex and zBMI.

**Table 2 children-10-01736-t002:** Changes in reported weight preoccupation and desire for body weight change (%) after a 6-week intervention.

			χ^2^, (*p*-Value)		
			Time	Group	Interaction
**Weight preoccupation**	PRE	POST			
**Total**					
Intervention (*n* = 156)	42	45	0.32, (0.57)	3.52, (0.06)	1.38, (0.24)
Control (*n* = 73)	56	46			
**Sex**					
**Boys**					
Intervention (*n* = 54)	33	32	0.06, (0.81)	2.71, (0.10)	0.00, (0.96)
Control (*n* = 33)	45	40			
**Girls**					
Intervention (*n* = 102)	47	52	0.34, (0.56)	1.13, 0.29	2.08, (0.15)
Control (*n* = 40)	65	50			
**Body weight status**					
**Underweight/Normal**					
Intervention (*n* = 106)	36	38	0.26, (0.61)	1.66, (0.20)	1.15, (0.28)
Control (*n* = 51)	49	39			
**Overweight/Obese**					
Intervention (*n* = 50)	56	59	0.13, (0.72)	1.16, (0.28)	0.42, (0.52)
Control (*n* = 22)	72	65			
**Desire for body weight change**					
**Total**					
Intervention (*n* = 162)	85	83	0.02, (0.89)	0.51, (0.48)	0.45, (0.50)
Control (*n* = 75)	84	87			
**Sex**					
**Boys**					
Intervention (*n* = 56)	80	73	0.00, (0.99)	1.09, (0.30)	1.17, (0.28)
Control (*n* = 33)	79	85			
**Girls**					
Intervention (*n* = 106)	88	89	0.01, (0.91)	0.01, (0.93)	0.01, (0.91)
Control (*n* = 42)	88	88			
**Body weight status**					
**Underweight/Normal**					
Intervention (*n* = 111)	84	80	0.05, (0.83)	0.00, (0.96)	0.40, (0.53)
Control (*n* = 53)	79	81			
**Overweight/Obese**					
Intervention (*n* = 51)	88	90	---	---	---
Control (*n* = 22)	95	100			

Note: Generalized linear mixed models were employed. The factors “Group” (intervention or control), “Time” (PRE or POST) and their interaction were treated as fixed effects. Participants and classes were treated as random effects to account for potential clustering. Covariates considered included sex (boys or girls) and body weight status (underweight/normal or overweight/obese). ---: absence of model convergence.

**Table 3 children-10-01736-t003:** Changes in reported body size misperception and dissatisfaction (%) after a 6-week intervention.

			χ^2^, (*p*-Value)		
			Time	Group	Interaction
**Body size misperception**	PRE	POST			
**Total**					
Intervention (*n* = 162)	86	88	0.13, (0.72)	1.84, (0.18)	0.13, (0.72)
Control (*n* = 75)	91	91			
**Sex**					
**Boys**					
Intervention (*n* = 56)	80	88	2.12, (0.15)	0.01, (0.93)	0.02, (0.89)
Control (*n* = 33)	79	88			
**Girls**					
Intervention (*n* = 106)	89	89	---	---	---
Control (*n* = 42)	100	93			
**Body weight status**					
**Underweight/Normal**					
Intervention (*n* = 111)	80	85	0.53, (0.47)	3.37, (0.07)	0.03, (0.87)
Control (*n* = 53)	89	91			
**Overweight/Obese**					
Intervention (*n* = 51)	98	96	0.72, (0.40)	0.37, (0.54)	0.00, (0.96)
Control (*n* = 22)	95	91			
**Body size dissatisfaction**					
**Total**					
Intervention (*n* = 162)	55	59	1.11, (0.29)	0.31, (0.58)	0.05, (0.82)
Control (*n* = 75)	56	61			
**Sex**					
**Boys**					
Intervention (*n* = 56)	52	59	2.01, (0.16)	1.96, (0.16)	0.22, (0.64)
Control (*n* = 33)	61	73			
**Girls**					
Intervention (*n* = 106)	57	58	0.03, (0.86)	0.81, (0.37)	0.03, (0.86)
Control (*n* = 42)	52	52			
**Body weight status**					
**Underweight/Normal**					
Intervention (*n* = 111)	46	51	1.88, (0.17)	0.82, (0.37)	0.15, (0.70)
Control (*n* = 53)	40	49			
**Overweight/Obese**					
Intervention (*n* = 51)	75	75	0.31, (0.58)	4.22, (0.04)	0.31, (0.58)
Control (*n* = 22)	95	91			

Note: Generalized linear mixed models were employed. The factors “Group” (intervention or control), “Time” (PRE or POST) and their interaction were treated as fixed effects. Participants and classes were treated as random effects to account for potential clustering. Covariates considered included sex (boys or girls) and body weight status (underweight/normal or overweight/obese). ---: absence of model convergence.

## Data Availability

Data will be available upon request.

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
