# Peer review of "Impact of a Web-Based Nutrition Intervention on Eating Behaviors and Body Size Preoccupations among Adolescents"

_children, 2023, doi:10.3390/children10111736_

Round 1
Reviewer 1 Report
· Abstract should start with some brief background information: A sentence or two giving a broad introduction to the study is needed. Please also include the time frame of the study/time period for data collection, the country where the study performed, and the statistical analysis used.
· Please make sure to define the study design when used the term "studies" in introduction and throughout (e.g., cross-sectional, longitudinal...etc.).
· The introduction generally reads well, is informative and provides a context for the study. However, authors have omitted literature on the topic (Int J Environ Res Public Health. 2019 May 14;16(10):1674; J Adolesc Health. 2011 Aug; 49(2): 148–154; Nutr J. 2022; 21: 75; PLoS One. 2017; 12(9): e0184509).
· Line 93-94 &102-103: It would also be useful to the reader to have an overview about these schools. How many schools? Were schools in big cities, small town or remote villages? Were they public or private? Were they classified as low, middle or high SES? How students were recruited?
· Line 112: Please change this to “Data collection”.
· How many dietitian or research professional teams (Line 131); and trained professionals (Line 141)?
· Line 151-159: The validity of self-administered questionnaires is unclear. Were these questionnaire piloted and standardized? What were the accuracy and consistency of these questionnaires? How the questionnaires were distributed?
· Supplementary figures are vague and unclear. It would be benefit to present mixed linear models and generalized linear mixed models in tables format.
· Authors should include hypothesis about the non-significant differences in the discussion.
· Line 328: The strengths of the study were not reported.
· Line 341-344: Information about future research directions are needed.
Reviewer 2 Report
Dear Authors,
This is an important application study for public health. The following minor comments are intended to strengthen the manuscript.
Materials and methods: There is no information about how many schools the invitation was sent to. This allows for the assessment of whether the sample size qualified for the study can be treated as a reliable representation. The number of potential participants is also important, i.e. the number of students aged 13-14 in a given region.
General remarks:
I might understand that the authors decided to combine underweight and normal weight respondents into one group. However, this should be indicated as a limitation.
Nutritional interventions usually focus on improving nutrition. Why do the authors emphasize in the discussion that this intervention had not produced a negative impact on nutritional traits? Could the authors comment on this?
Reviewer 3 Report
An interesting study assessing the impact of web-based school nutrition intervention on eating behavior traits and the subjective assessment of one's own weight and body shape by group of teenagers.
The introduction should show, what information about proper nutrition and body weight perception is provided to students as part of the regular school curriculum.
In subsection “Anthropometric measures” there are some inaccuracies.
In line 141-142 “Trained professionals at school measured adolescents’ height and weight according to standardized methods [24] at baseline (PRE, week 0) and after the intervention phase (POST, week 7)” but in line 148 – 149 we find “Anthropometric measurements were retaken after the intervention (post-1).” Please explain this content ?
Round 2
Reviewer 1 Report
Dear Authors,
I'm not convinced with using mixed models for repeated measures in figures format. I'll leave this to the associate editor but I believe tables would be more appropriate.
